# Structural basis for the non-self RNA-activated protease activity of the type III-E CRISPR nuclease-protease Craspase

Ning Cui[1,7], Jun-Tao Zhang[1,7], Zhuolin Li[1], Xiao-Yu Liu[1], Chongyuan Wang[2,3], Hongda Huang [4] ✉ & Ning Jia [1,5,6] ✉

The RNA-targeting type III-E CRISPR-gRAMP effector interacts with a caspase-like protease TPR-CHAT to form the CRISPR-guided caspase complex (Craspase), but their functional mechanism is unknown. Here, we report cryo-EM structures of the type III-E gRAMP[crRNA] and gRAMP[crRNA]-TPR-CHAT complexes, before and after either self or non-self RNA target binding, and elucidate the mechanisms underlying RNA-targeting and non-self RNA-induced protease activation. The associated TPR-CHAT adopted a distinct conformation upon self versus non-self RNA target binding, with nucleotides at positions −1 and −2 of the CRISPR-derived RNA (crRNA) serving as a sensor. Only binding of the non-self RNA target activated the TPR-CHAT protease, leading to cleavage of Csx30 protein. Furthermore, TPR-CHAT structurally resembled eukaryotic separase, but with a distinct mechanism for protease regulation. Our findings should facilitate the development of gRAMP-based RNA manipulation tools, and advance our understanding of the virus-host discrimination process governed by a nuclease-protease Craspase during type III-E CRISPR-Cas immunity.

Clustered regularly interspaced short palindromic repeats (CRISPR)/ CRISPR-associated protein (Cas) systems provide prokaryotes with adaptive immunity against invading viruses and plasmids[1,2]. The immunity is acquired by integrating the invading nucleic acids between repeat sequences of the host CRISPR locus. The CRISPR loci are then transcribed into precursor RNAs (pre-crRNAs) that are processed into mature CRISPR-derived RNAs (crRNAs), which assemble with single- and multi-subunit Cas proteins. The resulting crRNA-effector complexes detect and degrade the invading nucleic acids.

Depending on their Cas gene composition, CRISPR-Cas systems are classified into two major classes (1 and 2) and six types (I–VI)[3]. Class 1 systems (including types I, III, and IV) comprise multi-subunit effector complexes, whereas class 2 systems (including types II, V, and VI) are composed of single-subunit protein effectors. Of all known CRISPR-Cas systems, RNA-targeting effector complexes have been identified in type III, V, and VI systems[4–6], of which the single-subunit effector type VI Cas13 is widely used for RNA detection, editing, and knockdown[7]. However, upon recognition of the RNA targets, Cas13 exhibits non-specific RNA cleavage activity resulting in degradation of viral and host

[1]Department of Biochemistry, School of Medicine, Southern University of Science and Technology, Shenzhen 518055, China. [2]Faculty of Pharmaceutical Sciences, Shenzhen Institutes of Advanced Technology, Chinese Academy of Science, Shenzhen 518055, China. [3]Center for Human Tissues and Organs Degeneration, Shenzhen Institutes of Advanced Technology, Chinese Academy of Science, Shenzhen 518055, China. [4]Key Laboratory of Molecular Design for Plant Cell Factory of Guangdong Higher Education Institutes, Department of Biology, School of Life Sciences, Southern University of Science and Technology, Shenzhen 518055, China. [5]Shenzhen Key Laboratory of Cell Microenvironment, Guangdong Provincial Key Laboratory of Cell Microenvironment and Disease Research, Southern University of Science and Technology, Shenzhen 518055, China. [6]Key University Laboratory of Metabolism and Health of Guangdong, Southern University of Science and Technology, Shenzhen 518055, China. [7]These authors contributed equally: Ning Cui, Jun-Tao Zhang. ✉e-mail: huanghd@sustech.edu.cn; jian@sustech.edu.cn

RNA[5,8,9]. The nonspecific, collateral RNA cleavage activity of Cas13 is toxic to multiple mammalian cell types[10,11].

A recently characterized type III-E CRISPR-Cas subtype contains a single-protein effector that exhibits specific target RNA cleavage activity with no collateral activity or cell toxicity[3,12,13], representing a promising new tool for manipulating RNA. Kato et al. provided the first glimpse into the architecture of a type III-E CRISPR effector complex, *Desulfonema ishimotonii* Cas7-11, bound to a target RNA, elucidating the mechanism for target RNA cleavage[14]. However, given that the type III-E effector complex equally cleaves the invasive non-self and the host self RNA targets derived from the host CRISPR array[12], it is unknown if the type III-E effector complex discriminates self and non-self RNA targets during antiviral defense.

Type III CRISPR-Cas systems are divided into six subtypes (type III-A to F), of which type III-A, -D, -E, and -F contain Csm proteins, whereas type III-B and -C consist of Cmr proteins[3]. In previously reported canonical type III-A/B systems, the mature crRNA consists of a single spacer that is flanked by an 8-nt 5′-repeat tag derived from the repeat sequence of the host CRISPR array. The non-complementarity between 3′-flanking sequences of the target RNA and the 5′-repeat tag of crRNA designates an invasive non-self RNA target and triggers the immune response, whereas complementarity between these two sequence elements designates the RNA target derived from the host CRISPR array and prevents self-targeting[15]. Only binding of the invasive non-self RNA target, but not the host self RNA, allosterically activates the ssDNase and cyclic oligoadenylate (cOA) synthetase activities of the type III signature protein Cas10 to degrade the invading nucleic acids[16–20]. Meanwhile, cleavage of the invasive RNA target by the Csm3/Cmr4 subunit switches off Cas10 enzymatic activities, thereby preventing potential damage to the host due to persistent enzyme activities[18,21].

Lacking the type III signature protein Cas10, the III-E CRISPR loci often contain a gene encoding a caspase-like protease called TPR-CHAT that contains a caspase HetF associated with TPRs (CHAT) domain, which is a caspase family protease typically involved in programmed cell death, fused with a tetratricopeptide repeat (TPR) domain[3,12,13,22] (Fig. 1a). The TPR-CHAT protease interacts with *Candidatus* "Scalindua brodae" giant repeat-associated mysterious protein (gRAMP) to form the CRISPR-guided caspase complex (Craspase) in type III-E CRISPR-Cas systems[12], suggesting a functional relationship between CRISPR-Cas systems and caspase-like proteases during type III-E CRISPR-Cas antiviral immunity. The TPR-CHAT protease is implicated in regulating bacteria death by site-specific cleavage and subsequent activation of a bacterial gasdermin, which executes cell death for antiphage defense[23], suggesting an antiviral role of the TPR-CHAT protease in type III-E CRISPR-Cas immunity. However, we do not have a mechanistic understanding of the functional relationship between the type III-E CRISPR-Cas effector complex and the caspase-like protease TPR-CHAT, nor for the mechanism of the self versus non-self discrimination in type III-E systems.

Here, we combined structural biology and biochemistry analyses to generate high-resolution snapshots of the type III-E gRAMP^crRNA and gRAMP^crRNA-TPR-CHAT complexes, with or without their self or non-self RNA targets bound and revealed the mechanisms underlying RNA-targeting and virus-host discrimination governed by the non-self RNA-activated TPR-CHAT protease in type III-E CRISPR nuclease-protease Craspase-mediated antiviral immunity.

## Results

### Overall architecture of gRAMP^crRNA bound to RNA targets

The type III-E *Candidatus* "Scalindua brodae" gRAMP complex specifically recognizes and cleaves self and non-self RNA targets[12]. To investigate whether the gRAMP^crRNA complex can distinguish self from invasive non-self RNA targets, we first co-expressed the gRAMP gene with a single CRISPR array in *Escherichia coli* cells (Fig. 1a). gRAMP and

crRNA formed a stable complex (Supplementary Fig. 1a) and cleaved self RNA and non-self RNA targets into two products in a metal-dependent manner (Supplementary Fig. 1b). We then reconstructed the gRAMP^crRNA complexes with self or non-self RNA targets bound by the addition of ethylene diamine tetra acetic acid (EDTA) to prevent target cleavage (Supplementary Fig. 1c, d). We determined the cryogenic electron microscopy (cryo-EM) structures of gRAMP^crRNA-target RNA^self and gRAMP^crRNA-target RNA^non-self ternary complexes at 2.8 Å and 2.5 Å, respectively (Fig. 1c–i, Supplementary Figs. 2, 3), and observed greater densities in the head region of gRAMP^crRNA-target RNA^self.

The overall architecture of gRAMP^crRNA-target RNA ternary complex adopts a seahorse shape, resembling the type III-A CRISPR-Csm complex with fused homologous domains referred to as the head (the Csm5 domain), backbone (two Cas7-like Csm3 domains and one Cas11-like Csm2 domain) and tail (the Csm4 domain) fused by four linker domains (L1–L4), but lacking the type III signature protein Cas10[24–27] (Fig. 1b, f, i, Supplementary Fig. 4). We only observed densities corresponding to the Csm5 insertion domain in the head region in the gRAMP^crRNA-target RNA^self ternary complex, and only parts of the densities were well traced, suggesting its flexibility (Supplementary Fig. 2d). We observed a 36-nt crRNA bound to an 18-nt traceable segment of 56-nt non-self RNA target in gRAMP^crRNA-target RNA^non-self ternary complex (Fig. 1g, i). We observed more density for the crRNA-target RNA duplex in the gRAMP^crRNA-target RNA^self ternary complex containing a 42-nt crRNA bound to a 24-nt traceable self RNA target (Fig. 1c, f), with the extra observed densities corresponding to the anti-tag sequence that base-pairs with crRNA, and sequences covered by the Csm5 insertion domain. Despite the difference between the self and invasive non-self RNA targets, we observed minimal conformational changes in the overall structure between gRAMP^crRNA-target RNA^non-self and gRAMP^crRNA-target RNA^self, with a small root-mean-square deviation (r.m.s.d.) of 0.463 Å (Fig. 1j), indicating the self versus non-self discrimination does not result from conformational changes of gRAMP^crRNA induced by binding of self or non-self RNA targets.

### Assembly of mature crRNA

In the canonical type III systems, the mature crRNA with an 8-nt 5′-repeat tag is produced from pre-crRNA following the cleavage by a stand-alone endoribonuclease Cas6[2,28]. The Cas7.1 domain in type III-E *D. ishimotonii* Cas7-11 (corresponding to the Csm4 domain in gRAMP) is responsible for pre-crRNA processing to generate the mature crRNA with a 15-nt 5′-repeat tag[13,14]. However, we observed an 18-nt 5′-repeat tag in our type III-E gRAMP complexes, numbered −18 to −1 according to the convention (Fig. 2a), with no cleavage observed between nucleotides −15U and −16C. In addition, the indispensable catalytic residue H43 adjacent to the scissile phosphate at the −15 to −16 step in *D. ishimotonii* Cas7-11 for pre-crRNA processing was replaced by T45 in gRAMP (Fig. 2b, blue inset, Supplementary Fig. 5a), implying a different pre-crRNA processing mechanism for gRAMP.

The nucleotides at positions −1 to −16 in the 18-nt 5′-repeat tag formed multiple sequence-specific contacts with residues in Csm4 and Csm3 (Supplementary Fig. 5b), indicating gRAMP recognizes the 5′-repeat tag in a sequence-dependent manner, consistent with the conservation of the 5′-repeat tag sequence (Supplementary Fig. 5c). In contrast, we found a series of sequence-nonspecific contacts between the crRNA spacer sequence and Csm3.1, Csm3.2, and Csm5 (Supplementary Fig. 5b), accounting for the lack of sequence specificity for spacer sequence recognition. Notably, a metal ion was coordinated by the phosphate group of −11U, the side chain of D137, and the carboxyl group of G134 and A139 in the gRAMP complex (Fig. 2b, blue inset, Supplementary Fig. 5a). A magnesium ion critical for guide RNA stabilization is located in a similar position in prokaryotic Argonautes[29], suggesting the metal ion observed here may also function in stabilizing the 5′-repeat tag in the gRAMP complex.

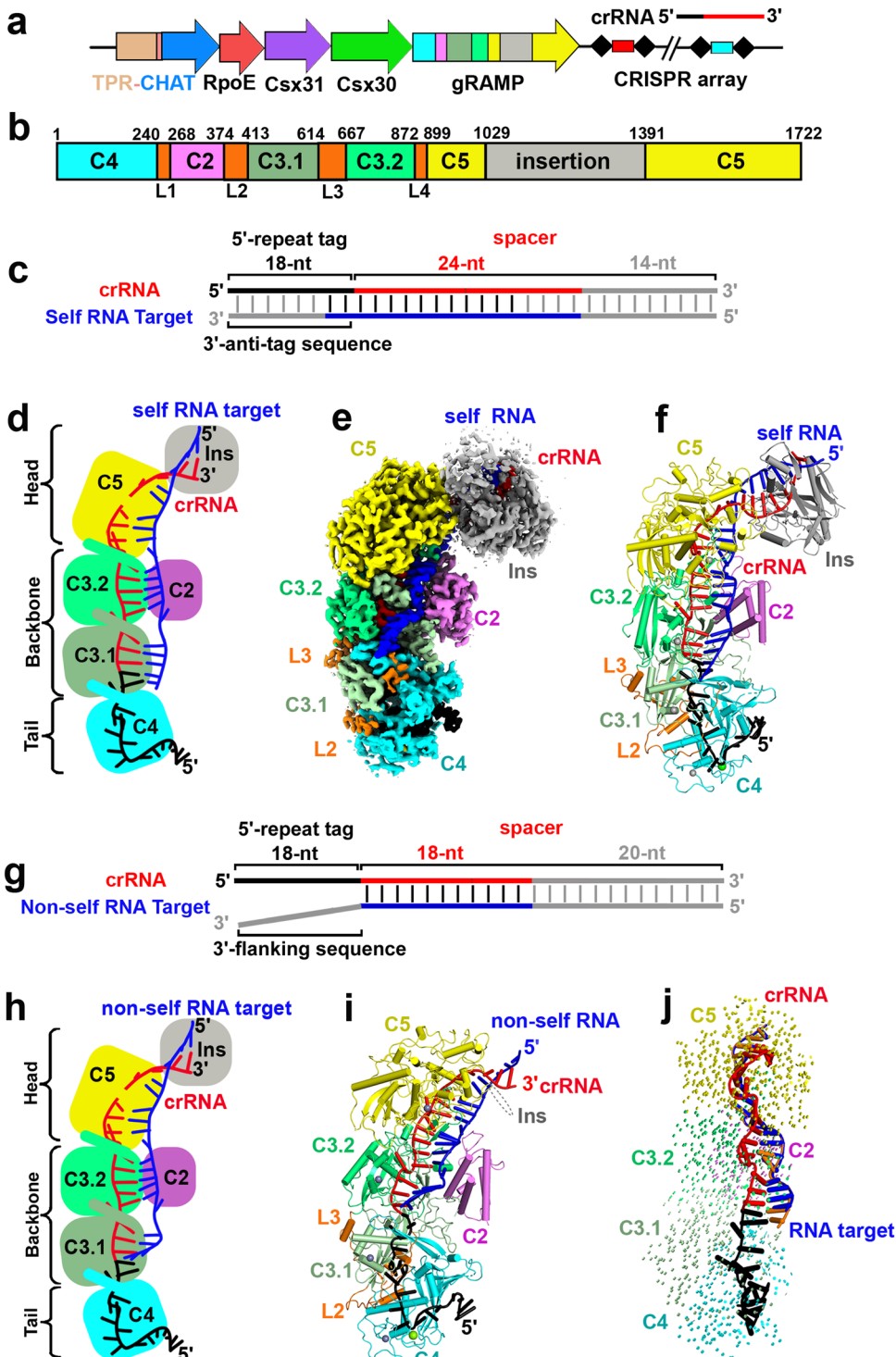

**Fig. 1 | Cryo-EM structures of gRAMP^crRNA bound to target RNAs. a** The outline of type III-E CRISPR-Cas loci from *Candidatus* "Scalindua brodae". gRAMP and TPR-CHAT genes are colored according to the domain organization. The CRISPR locus is composed of the host nucleotide repeats (black diamonds) separated by invading spacer sequences (colored cylinders). **b** Domain organization of gRAMP protein. C2, Csm2 domain; L1, Loop 1; insertion indicates the insertion domain within Csm5 domain; Other domains are named accordingly. **c** Schematic representation of pairing between crRNA and self RNA target. Segments that can be traced in gRAMP^crRNA-target RNA^self complex are in color, while disordered segments are in gray. Schematic (**d**), cryo-EM reconstruction (**e**) and ribbon representation (**f**) of the 2.8 Å structure of gRAMP^crRNA-target RNA^self complex. The gray spheres indicate the zinc ions bound in the gRAMP complex. **g** Schematic representation of pairing between crRNA and non-self RNA target. Segments that can be traced in gRAMP^crRNA-target RNA^non-self complex are in color, while disordered segments are in gray. Schematic (**h**), and ribbon representation (**i**) of the 2.5 Å structure of gRAMP^crRNA-target RNA^non-self complex. **j** Structural comparison between gRAMP^crRNA-target RNA^non-self and gRAMP^crRNA-target RNA^self complexes. Vector length correlates with the domain movement scale.

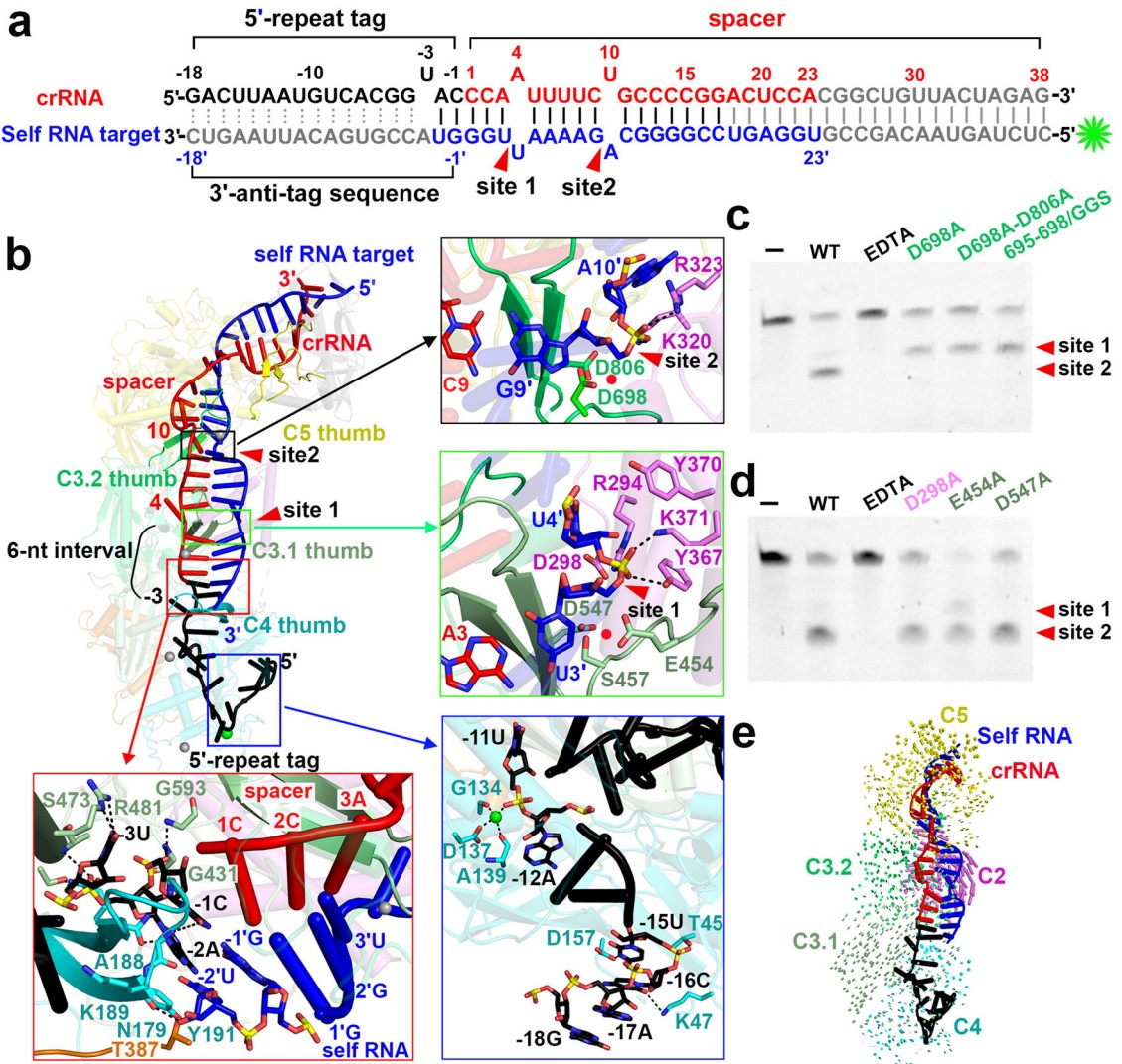

**Fig. 2 | Assembly of crRNA-Target RNA Duplex in gRAMP^crRNA-Target RNA^self complex. a** Schematic drawing for the sequences of crRNA and self RNA target. Potential cleavage sites are mapped on the target RNA sequence. Segments that can be traced are in color, while disordered segments are in gray. The green asterisk indicates the fluorescence label at 5′ end of the RNA target for the following target RNA cleavage. **b** The thumb elements of Csm4 and Csm3 domains kink the crRNA-target RNA duplex every 6th nucleotide, with no kink associated with the thumb of Csm5 domain. The black and green insets present interactions that might contribute to RNase activity for the target RNA cleavage at the 9′–10′ step and 3′–4′ step, respectively. The blue inset provides detailed contacts between Csm4 domain and nucleotides at 5′-repeat tag. The green sphere indicates a metal ion. The red inset shows detailed interactions between protein residues and crRNA-target RNA duplex at position −1 and −2. **c** and **d** Cleavage of target RNA by Csm3.2 (**c**) and Csm3.1 (**d**) mutants in the context of the gRAMP^crRNA-target RNA complex. The two cleavage sites are indicated as red arrows. In vitro RNA cleavage experiments were repeated at least three times with similar results. **e** Structure comparison between gRAMP^crRNA binary complex and gRAMP^crRNA -target RNA^self ternary complex based on alignment of the Csm4 domain. Vector length correlates with the domain movement scale.

## Metal-dependent-specific cleavage of target RNA

The crRNA guides gRAMP to cleave self and non-self RNA targets between nucleotides 3′ and 4′ (site 1) and between nucleotides 9′ and 10′ (site 2) with a 6-nt interval in a metal-dependent manner[12] (Fig. 2a, Supplementary Fig. 1b). The potential position of a divalent cation indispensable for target RNA cleavage is indicated by the red ball depicted among the acidic Asp side chains in Csm3.1 and Csm3.2 domains and the phosphate oxygen (Fig. 2b, black and green insets, Supplementary Fig. 5d, e). Consistently, mutation of the respective acidic Asp residues (D698 in Csm3.2, and D547 in Csm3.1), which may coordinate the indispensable metal ion, into alanine abolished the RNase activity in Csm3.1 and Csm3.2 domains (Fig. 2c, d). Notably, mutation of residue D298 in Csm2 into alanine also abolished RNase activity (Fig. 2d), indicating the critical role of Csm2 for target RNA cleavage.

In addition, the two cleavable phosphates were stabilized by residues in the Csm2 domain (Fig. 2b, black and green insets, Supplementary Fig. 5d, e). Notably, these residues undergo substantial conformational changes revealed by the structural comparison between the gRAMP^crRNA-target RNA ternary complex and the gRAMP^crRNA binary, whose structure we determined at 2.7 Å (Fig. 2e, Supplementary Figs. 6 and 7a).

The mechanism of target RNA cleavage by gRAMP^crRNA is reminiscent of previously reported type III multi-subunit effector complexes[30–32], indicating type III-E gRAMP retains a similar RNA-targeting mechanism even after domain fusion into a single-protein effector. Previous reports have implicated that the deprotonation of the 2′-hydroxyl group of target RNA initiates the nucleophilic attack of the scissile bond, leading to the generation of fragments with 3′-phosphate (or 2′, 3′-cyclic phosphate) and 5′-hydroxyl termini[6,33]. Thus,

we speculate a similar catalytic pathway for target RNA cleavage by type III-E gRAMP complexes. We observed that the 2'-hydroxyl group of A4' and A10' are close to the basic residues K371 and K320 (Fig. 2b, green and black insets), which could work as a catalytic base by deprotonating the 2'-hydroxyl group of A4' and A10', respectively. The deprotonation of the 2'-hydroxyl group facilitates the nucleophilic attack on the scissile bond, resulting in the formation of the products with 3'-phosphate (or 2', 3'-cyclic phosphate) and 5'-hydroxyl termini. The indispensable acidic residues (D298, D547, and D698) are positioned 4.5–8.5 Å away from the scissile bond (Fig. 2b, green and black insets), and might contribute to stabilize the indispensable $Mg^{2+}$, which in turn facilitates the cleavage chemistry in the active gRAMP complexes.

Despite the similarity, a kink at position −3 in the gRAMP$^{crRNA}$ complex rather than a kink at position −1 in the canonical type III Csm/Cmr complexes[24,25,27,32] separates nucleotides between the 5'-repeat tag and the spacer sequence of crRNA, acting as a start site to measure the 6-nt periodic RNA cleavage sites (Fig. 2a, b). In canonical type III Csm/Cmr complexes, nucleotides at positions −2 to −5 within the 5'-repeat tag form a stacked alignment in a pseudo A-form configuration and are exposed to the solvent, serving as sensors to avoid autoimmunity[24–26,32]. By contrast, in the type III-E gRAMP$^{crRNA}$ complex, nucleotides −1C and −2A within the 5'-repeat tag stack with 1C to 3A within the spacer, which are available for base pairing with the self RNA target (Fig. 2a, b, red inset), possibly acting as sensors to discriminate self and non-self RNA. However, the gRAMP$^{crRNA}$ complex equally cleaved self and non-self RNA targets (Supplementary Fig. 1b) with minimal differences observed in RNA catalytic pockets between the self and non-self RNA-bound gRAMP$^{crRNA}$ complex (Supplementary Fig. 7b), suggesting factors other than RNA cleavage are responsible for the self and non-self-discrimination in type III-E CRISPR antiviral immunity.

## The caspase-like protease TPR-CHAT binds to the tail of the gRAMP$^{crRNA}$ complex

In canonical type III CRISPR-Cas systems, self and non-self RNA are equally cleaved by Csm3 subunits, but only the invasive non-self RNA activates the ssDNase and cOA synthetase activities of the signature Cas10 subunit, which is critical for type III antiviral immunity[16–20]. Given that it lacked the type III signature *cas10* gene, the type III-E CRISPR loci contain a gene encoding a caspase-like protease TPR-CHAT (Fig. 1a), which formed a stable complex with gRAMP$^{crRNA}$ (Supplementary Fig. 8a). To explore how the TPR-CHAT protease interacts with the gRAMP$^{crRNA}$ complex, we determined the cryo-EM structure of the gRAMP$^{crRNA}$-TPR-CHAT complex at 2.6 Å resolution (Fig. 3a, b, Supplementary Fig. 8b–e). TPR-CHAT contained an N-terminal TPR domain, and a caspase-like protease CHAT domain connected by a linker domain and bound the tail of the gRAMP$^{crRNA}$ complex mainly through Loop2-mediated electrostatic contacts with a buried interface area of ~1800 Å$^2$ (Fig. 3c, Supplementary Fig. 8f).

## The structure of bacterial TPR-CHAT resembles that of eukaryotic separase

A DALI structural comparison revealed that TPR-CHAT structurally resembles eukaryotic separase, which corroborates previous bioinformatic analysis that the bacterial CHAT domain is most closely related to the eukaryotic separase, a caspase-like cysteine protease essential for cohesion dissolution during chromosome segregation[34]. In addition, the bacterial TPR-CHAT contained the highly conserved histidine-cysteine catalytic dyad typical of caspase-like proteases across bacteria and humans (Fig. 3d). In eukaryotes, separase activity is tightly regulated, otherwise it would lead to missegregation and aneuploidy, thereby resulting in birth defects and cancer[35–37]. Separase activity is typically inhibited by forming a stable complex with securin, while degrading the inhibitory securing activates separase[37].

The structural similarity between bacterial TPR-CHAT and human separase, including the highly conserved catalytic dyad H585 and C627 (corresponding to H2003 and C2029 in human separase) (Fig. 3e, f), suggests the potential protease activity of bacterial TPR-CHAT. Bacterial TPR-CHAT may cleave and activate the bacterial cell death effector gasdermin to trigger cell death during anti-phage defense[23], which implicates the bacterial TPR-CHAT protease in preventing bacteriophage propagation by triggering host suicide, thereby requiring that its activity also be strictly regulated in bacteria. Notably, as TPR-CHAT lacks an inhibitory peptide, the catalytic C627 and H585 residues were located in a rigid α helix and a β sheet, named C-helix and H-sheet, respectively, and buried by their own residues (D543, T629, and D630) via electrostatic contacts, making the catalytic residues inaccessible to incoming substrates. In addition, the distance between H585 and C627 in bacterial TPR-CHAT was ~7 Å (Fig. 3e, inset), similar to the distance between the H2003 and C2029 in the inactive human separase (Fig. 3f, inset), the activity of which is inhibited by forming a complex with the inhibitory securin[37], suggesting the TPR-CHAT adopts an inactive state. Taking these observations together, we speculate that the TPR-CHAT protease adopts an autoinhibitory inactive conformation by forming a stable complex with gRAMP$^{crRNA}$, reminiscent of the autoinhibited ssDNase activity of the Cas10 subunit in the type III-A Csm$^{crRNA}$ complex[24,25,32].

## Binding of the invasive non-self RNA target induces structural rearrangement of TPR-CHAT

To determine if the TPR-CHAT protease is activated during phage infection, we investigated the conformational changes of TPR-CHAT upon binding of the invasive non-self RNA target to the gRAMP$^{crRNA}$-TPR-CHAT complex. Then, we determined a 2.9 Å cryo-EM structure of an invasive non-self RNA target-bound gRAMP$^{crRNA}$-TPR-CHAT quaternary complex (Fig. 4a–c; Supplementary Fig. 9a–d). The overall structure adopted a similar architecture to the gRAMP$^{crRNA}$-TPR-CHAT ternary complex with an r.m.s.d of 1.78 Å, containing a 38-nt crRNA a`nd a 23-nt traceable segment of the 56-nt non-self RNA target (Fig. 4a), but with a significant conformational rearrangement of the linker and CHAT protease domains of TPR-CHAT by as much as ~26 Å (Fig. 4d).

Notably, the 3'-flanking sequence of the invasive non-self RNA target at position −5' to −1' was non-complementary to the 5'-repeat tag of the crRNA (Fig. 4a), swinging away from the crRNA and directly interacting with TPR-CHAT, where it resided in a cleft between the linker and TPR domains (Fig. 4b, c, Supplementary Fig. 10). Nucleotide-1' C was stabilized by K187$^{Csm4}$ and R382$^{Csm3.1}$, while the nucleotides at positions −2' to −5' directly interreacted with TPR-CHAT mainly through sequence-nonspecific contacts, which induced large conformational changes of the linker and the CHAT protease catalytic pocket (Fig. 4d). In particular, nucleotides −2' A and −4' G directly interacted with Y360 and V361 of the linker domain, inducing dramatic conformational changes of the linker domain followed by rearrangement of the H-sheet and C-helix into the extended loop structures in the CHAT protease catalytic pocket (Fig. 4e). The conformational changes in the catalytic pocket resulted in outward movement and exposure of the catalytic residues H585 and C627, which became accessible to the potential substrate peptide (in marine, Fig. 4e, f), compared to the structure before the non-self RNA target binding (in gray, Fig. 4e, g). Exposure of the catalytic cysteine to the solvent and reorganization of the C-loop indicate activation of the caspase-like protease[38,39], suggesting invasive non-self RNA target binding activates the TPR-CHAT protease. Most importantly, the distance between the catalytic dyad H585 and C627 changed from ~7 Å in the inactive gRAMP$^{crRNA}$-TPR-CHAT complex (in gray) to ~3 Å (in marine) upon non-self RNA target binding, a distance at which both of the catalytic residues were positioned for attack on the peptide bond of the substrate (Fig. 4e). Furthermore, upon target binding of the invasive non-self

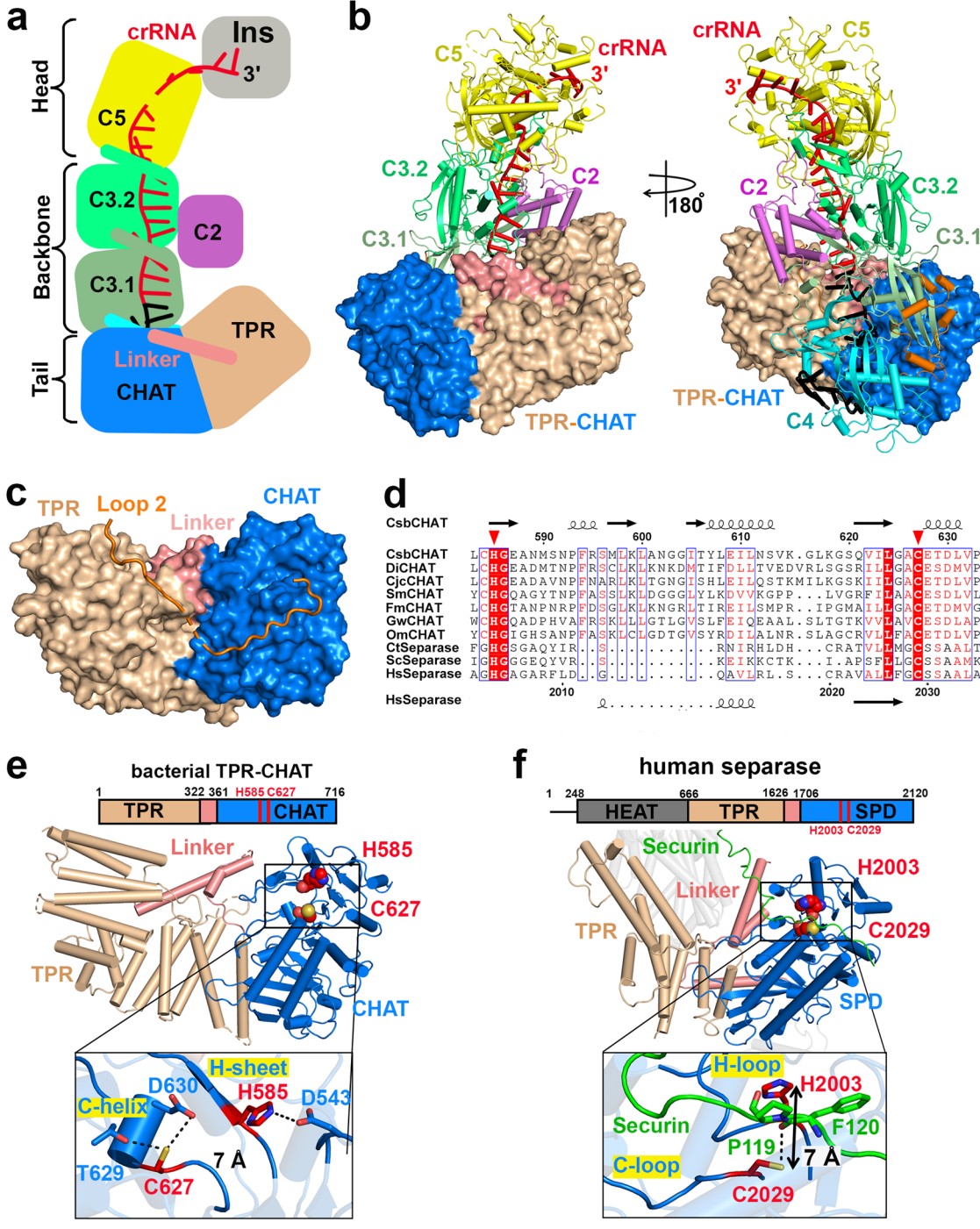

**Fig. 3 | Cryo-EM structure of gRAMP^crRNA in complex with TPR-CHAT.** Schematic (**a**) and ribbon (**b**) representations of cryo-EM structure of gRAMP^crRNA-TPR-CHAT complex at 2.6 Å resolution. The associate TPR-CHAT subunit is shown in surface view. **c** TPR-CHAT binds to Loop 2 in gRAMP. **d** Multiple sequence alignment of representative CHAT and separase orthologues. The conserved catalytic residues H585 and C627 are indicated by red triangles. CsbCHAT: KHE91663.1; DiCHAT: WP_124327588.1; CjcCHAT: KAA0249747.1; SmCHAT: OBJA01001127.1; FmCHAT:

SESD01000293.1; GwCHAT: MGTA01000040.1; OmCHAT: PDWI01005922.1; Ct: *Chaetomium thermophilum*; Sc: *Saccharomyces cerevisiae*; Hs: *Homo sapiens*. Structures of bacterial TPR-CHAT (**e**) and human separase (PDB 7NJ1) (**f**) with an expanded view of the catalytic pocket. The catalytic residues are shown as red spheres. TPR, Tetratricopeptide Repeat Domain; CHAT, Caspase HetF Associated with TPRs; SPD, Separase Protease Domain.

RNA, the C-loop conformation and the catalytic dyad H585 and C627 geometry switched from an inactive state (in gray) to a state (in marine) similar to the active *Chaetomium thermophilum* separase (in violet)[35] (Fig. 4h), suggesting that invasive non-self RNA binding allosterically activates the TPR-CHAT protease in the gRAMP^crRNA-TPR-CHAT complex.

## The 3'-anti-tag of self RNA binds to a channel distinct from the 3'-flanking sequence of non-self RNA

If binding of the invasive non-self RNA activates the TPR-CHAT protease, which might be involved in host suicide, binding of self RNA should not activate the TPR-CHAT protease to prevent self-targeting. To test this hypothesis, we determined a 2.7 Å cryo-EM structure of

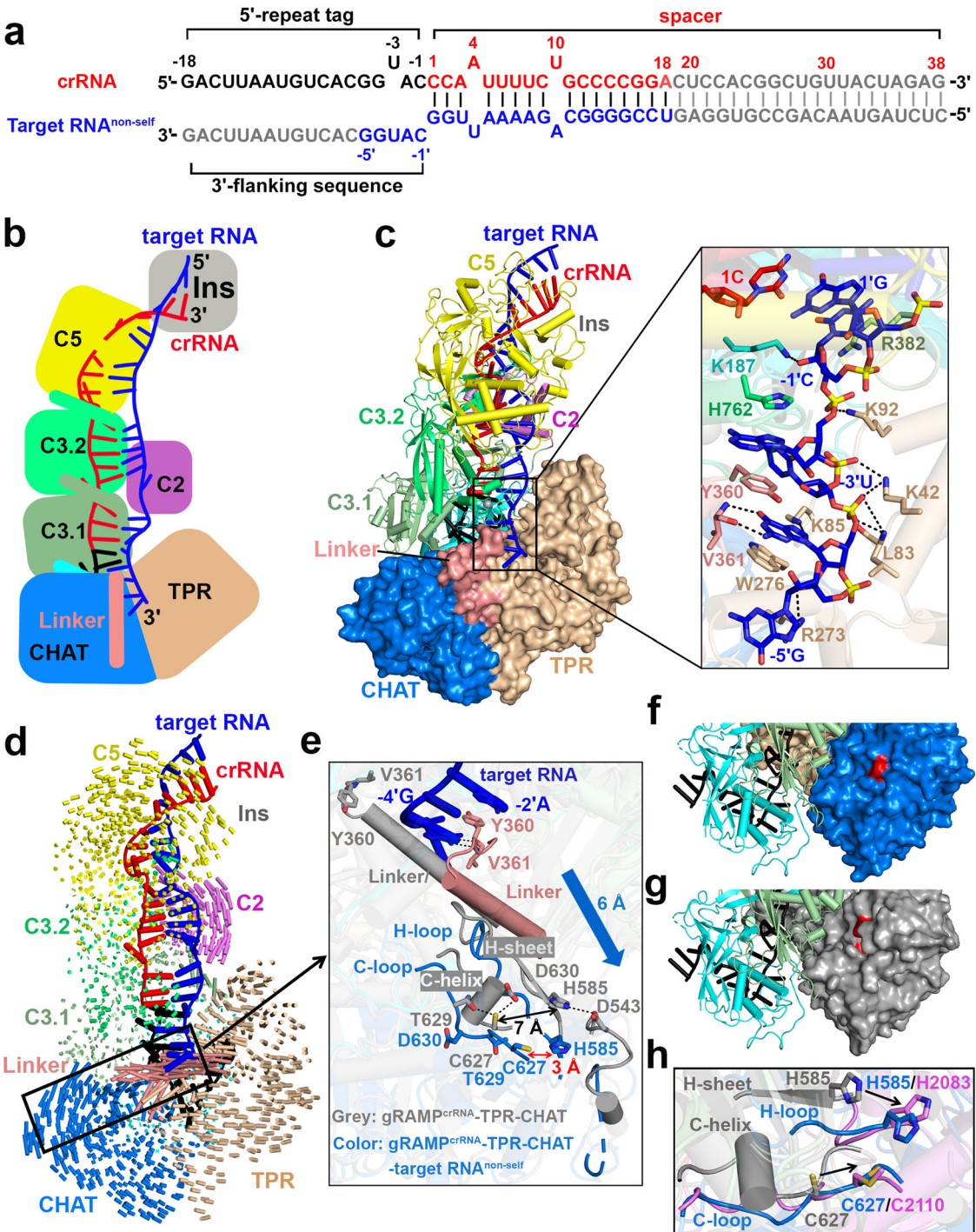

**Fig. 4 | Binding of invasive non-self RNA target induces large conformational changes of gRAMP^crRNA -TPR-CHAT complex. a** Schematic drawing for the sequences of crRNA and non-self RNA target. Segments that can be traced are in color, while disordered segments are in gray. Schematic (**b**) and ribbon (**c**) representations of cryo-EM structure of gRAMP^crRNA-TPR-CHAT complex bound to non-self RNA target. TPR-CHAT protein is shown in a surface view. The expanded view shows the detailed interactions between 3′- flanking sequence and protein residues in gRAMP and TPR-CHAT. **d** Structural comparison between gRAMP^crRNA-TPR-CHAT complex before and after binding to non-self RNA target. Vector length correlates with the domain movement scale. **e** Detailed conformational changes induced by

non-self target RNA binding. The gRAMP^crRNA-TPR-CHAT complex is shown in gray, and gRAMP^crRNA-TPR-CHAT-target RNA^non-self complex is shown in color. The blue arrow indicates the movement scale of the secondary structures in the CHAT protease catalytic pocket. Structures of CHAT protease domain in gRAMP^crRNA -TPR-CHAT before (**g**) and after (**f**) non-self RNA target binding. The CHAT domain is shown in a surface view. The red surface indicates the position of catalytic residues H585 and C627. **h** Structural comparison of the catalytic pockets among gRAMP^crRNA-TPR-CHAT (in gray), gRAMP^crRNA-TPR-CHAT-target RNA^non-self (in marine), and the active separase protease domain from the thermophilic fungus *Chaetomium thermophilum* (in violet, PDB 5FC3).

gRAMP^crRNA-TPR-CHAT bound to a self-RNA target with an anti-tag sequence complementary to the 5′-repeat tag (Fig. 5a–c; Supplementary Fig. 9e–h). Superposition of structures between bound self and non-self RNA targets reveals pronounced conformational differences in the positions of the linker and the CHAT protease domain (Fig. 5d), indicating that the gRAMP^crRNA-TPR-CHAT complex distinguishes self from non-self RNA targets, which is manifested in the distinct conformation of TPR-CHAT.

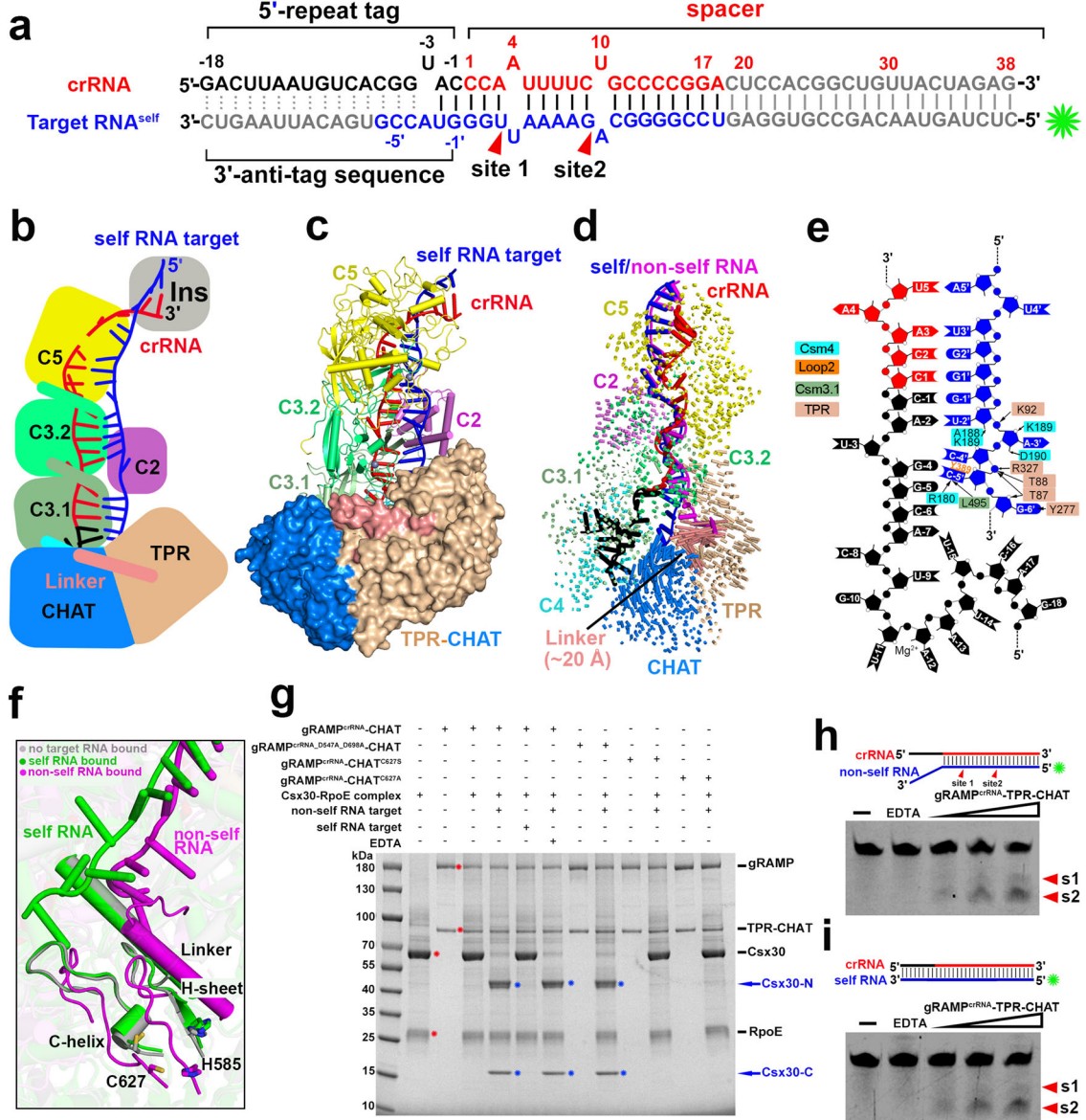

**Fig. 5 | Cryo-EM structure of gRAMP^crRNA-TPR-CHAT bound to self RNA target.**
**a** Schematic representation for the sequences of crRNA and self RNA target. The traced and disordered segments are in color and gray, respectively. Schematic (**b**) and cartoon (**c**) representations of cryo-EM structure of self RNA target bound to gRAMP^crRNA-TPR-CHAT quaternary complex. TPR-CHAT protein is shown in a surface view. **d** Structural comparison between gRAMP^crRNA-TPR-CHAT complex bound to self and non-self RNA targets. Vector length correlates with the domain movement scale. **e** Schematic interactions between the 3'-flanking sequence and protein residues. Hydrogen bonds and stacking interactions are indicated by black arrows and orange lines, respectively. **f** Conformational changes of the Linker and CHAT protease catalytic pocket in TPR-CHAT induced by self (in green) and non-

self (in violet) RNA target binding. The TPR-CHAT subunit before target RNA binding is colored in gray. **g** Cleavage of Csx30-RpoE complex by the non-self RNA target activated protease activity of gRAMP^crRNA-TPR-CHAT complex. The red and blue asterisks indicate the position of input proteins and cleaved products, respectively. Cleavage of non-self (**h**) and self (**i**) RNA targets by gRAMP^crRNA-TPR-CHAT complex. Increasing concentrations (100, 200, 400 nM) of gRAMP^crRNA-TPR-CHAT complexes were incubated with 50 nM 5' 6-FAM-labeled non-self or self RNA targets at 37 °C for 60 min, respectively. The red arrows indicate the cleavage sites. The green asterisk indicates the fluorescence label at 5' end of target RNA. In vitro cleavage assays were repeated at least three times with similar results.

Notably, distinct from the 3'-flanking sequence of non-self RNA that resides in the cleft between the linker and the TPR domain in TPR-CHAT (Fig. 4c), the 3'-anti-tag sequence of self RNA bound in a channel between TPR-CHAT and gRAMP (Fig. 5c). Nucleotides −1'G and −2'U of the self RNA base paired with nucleotides −1C and −2A within the 5'-repeat tag of the crRNA, and these base-pairing contacts forced the other anti-tag sequence at position −3' to −6' to bind in a cleft between Csm4 and the TPR domain via sequence-specific and nonspecific contacts (Fig. 5e). This binding mode of the 3'-anti-tag sequence of the self RNA induces minimal conformational changes of the gRAMP^crRNA-TPR-CHAT complex, including the C-helix and H-sheet conformation

and the catalytic dyad H585 and C627 geometry in the CHAT protease domain (Fig. 5f, Supplementary Fig. 11, 12a), which contrasts with those induced by the 3'-flanking sequence of non-self RNA (Figs. 4d, 5f), suggesting that the TPR-CHAT protease adopts an autoinhibitory conformation upon own self RNA target binding, thereby preventing self-targeting to avoid autoimmunity.

**Foreign non-self RNA activates the TPR-CHAT protease of type III-E Craspase**
To test if only the foreign non-self RNA target activates the TPR-CHAT protease of the type III-E Craspase in vitro, we first searched for

**TPR-CHAT protease substrates.** The substrates of CHAT proteases are commonly encoded by genes located near the protease-domain containing genes[23]. In type III-E CRISPR loci (Fig. 1a), the genes encoding the proteins Csx30 and Csx31 remain mysterious, with the other gene known to encode the sigma factor RpoE. We wondered if RpoE, Csx30, or Csx31 are TPR-CHAT substrates. We coexpressed these three components together in *E. coli* cells. The gel filtration assay showed RpoE formed a stable complex with Csx30 (Supplementary Fig. 12b). Only Csx30 within the Csx30-RpoE complex was cleaved into two separate products by gRAMP[crRNA]-TPR-CHAT complexes upon non-self RNA target binding but not self RNA target binding (Fig. 5g), which is consistent with our structural analysis. The nickel affinity chromatography assay showed that Csx30 was cleaved into an N-terminal product with high molecular weight (Csx30-N) and C-terminal product with low molecular weight (Csx30-C). Csx30-N still formed a stable complex with RpoE, while Csx30-C was released from the Csx30-RpoE complex after cleavage of Csx30 (Supplementary Fig. 12c). The gel filtration chromatography assay further confirmed that RpoE was still associated with Csx30-N after cleavage of Csx30, given that the cleavage product Csx30-N was eluted with RpoE in a single peak (Supplementary Fig. 12d). We then purified the individual Csx30 and Csx31, of which only Csx30 was cleaved by the non-self RNA-activated TPR-CHAT protease (Supplementary Fig. 12e–g), further confirming that Csx30 is the substrate of TPR-CHAT protease.

Notably, mutation of the catalytic residue C627 in the TPR-CHAT protease into alanine or serine abolished its protease activity (Fig. 5g, Supplementary Fig. 12e, g), demonstrating the catalytic pocket in the TPR-CHAT protease. Neither mutation of the RNase catalytic residues nor addition of EDTA, which suppressed the RNase activities of the Csm3 domains (Fig. 2c, d), affected TPR-CHAT protease activities in gRAMP[crRNA]-TPR-CHAT complexes (Fig. 5g). Further, non-self and self RNA targets were recognized and cleaved by Csm3 subunits in the gRAMP[crRNA]-TPR-CHAT complex (Fig. 5h, i), suggesting that the RNase activity temporally controls CHAT protease activity, thereby preventing potential damage to host cells caused by continuous enzymatic activity. Thus, the base-pairing potential between crRNA and target RNA at positions −1 and −2, rather than RNase activity, is responsible for virus–host discrimination, and switches the activity of the TPR-CHAT protease within the type III-E Craspase on or off, providing a regulatory mechanism for the activity of the caspase-like protease distinct from its eukaryotic homolog separase.

## Discussion

Type III-A CRISPR-Cas immunity against invading bacteriophages and plasmids is achieved by a crRNA-bound multi-subunit Csm complex (Csm2 to Csm5, and Cas10 subunits). This complex binds to the invasive non-self RNA target, which activates the ssDNase and cOA synthetase of the signature protein Cas10[16–20] (Fig. 6a, b). The second messenger cOA allosterically activates the non-specific RNase Csm6, which provides the host with further protection against invading phages and plasmids[40,41]. Both enzymic activities of Cas10 are switched off by sequence-specific cleavage of the invasive RNA targets by the RNase Csm3 subunits, providing temporal control of Cas10 activities and thereby preventing potential damage to host cells[18,21]. The base-pairing potential between elements of the 5′-repeat tag at position −2 to −5 and the 3′-sequence of target RNA is responsible for discriminating self from non-self RNA targets[2,18].

The type III-E CRISPR-Cas system contains a single-protein effector, demonstrating specific recognition and cleavage of target RNA without collateral activity and cell toxicity[12,13], representing a promising new RNA manipulation tool. While the type III-E CRISPR-Cas system lacks the signature Cas10 protein crucial for canonical type III CRISPR immunity[16–20], type III-E CRISPR loci encode a caspase-like protease, TPR-CHAT, which directly interacts with the type III-E effector complex gRAMP to form Craspase[12], and TPR-CHAT protease resides in a similar position as Cas10 in the type III-A effector complex[24–27] (Fig. 6a, c). TPR-CHAT is implicated in regulating cell death during antiphage defense[23], suggesting that the type III-E CRISPR-Cas systems defend against viral infection by Craspase-triggered host suicide. Using structural biology and biochemistry approaches, we uncovered the structural basis for the functional relationship between the caspase-like protease and CRISPR-Cas systems, leading to a proposed model for the non-self target RNA-activated TPR-CHAT protease to confer type III-E antiviral immunity, suggesting a neofunctionalization of type III-E CRISPR–Cas systems (Fig. 6d).

The TPR-CHAT protease stays in an inactive autoinhibitory state by forming a stable complex with gRAMP[crRNA]. Upon binding of invasive non-self RNA targets, its 3′-flanking sequence induces conformational changes of the TPR-CHAT linker domain by as much as ~20 Å followed by rearrangement of the CHAT protease catalytic pocket, thereby activating the CHAT protease that cleaves the protein substrate Csx30. The autoinhibitory rigid C-helix and H-sheet in the catalytic pocket are transformed into flexible C-loop and H-loop elements and the catalytic dyad H585 and C627 become accessible to attack the peptide bond of the substrate (Fig. 4e, h). The subsequent cleavage of the RNA targets by Csm3 domains releases the RNA targets, thereby switching off TPR-CHAT protease activity (Fig. 5g, h). By contrast, to bind the self RNA target, the 3′-anti-tag sequence takes a distinct route from the 3′-flanking sequence of the non-self RNA target, while inducing minimal conformational changes of the TPR-CHAT protease (Fig. 5f, Supplementary Fig. 12a), which remains in an autoinhibitory state. Thus, the base-pairing potential between elements of the 5′-repeat tag of crRNA and the 3′-sequence of the RNA target at positions −1 and −2 plays a key role in discriminating self from non-self RNA (Figs. 4a, 5a). Furthermore, the RNase activity of Csm3 domains may temporally control TPR-CHAT protease activity, thereby preventing potential damage to host cells.

Interesting questions that remain to be addressed are whether and how the cleaved Csx30 induces bacterial cell death, given that the TPR-CHAT protease cleaves and activates the bacterial cell death effector gasdermin, triggering cell death during antiphage defense[23]. Structural studies established homology between bacterial TPR-CHAT protease and eukaryotic separase, which is known to cleave after the conserved EXXR (X represents any residue) peptide motif[35]. However, the critical separase residue D2151 responsible for the conserved R215 recognition at the P1 position is replaced by K670 in bacterial TPR-CHAT, and the residues around E212 at P4 position is also different between eukaryotic separase and bacterial TPR-CHAT (Supplementary Fig. 13), suggesting a distinct substrate specificity of TPR-CHAT from that for eukaryotic separase[35]. Recent studies reported that the cleavage site of Csx30 is after residue L407[42,43], and the cleavage of Csx30 involves in cell death[43–45]. Further, we also found Csx30 formed a stable complex with RpoE (Supplementary Fig. 12b), and the cleavage product Csx30-N was stably associated with RpoE (Supplementary Fig. 12c and d). A recent study has shown that the RpoE transcriptional activity, which might be involved in spacer acquisition or other immune response, is regulated by the binding and follwing cleavage of Csx30[45]. It will be interesting to further investigate the molecular mechanisms underlying Csx30 cleavage by the TPR-CHAT protease and its potential involvement in the Craspase-mediated defense pathways. Nevertheless, we elucidated the RNA-targeting mechanisms of the gRAMP[crRNA] complex, providing a structural basis for developing new promising gRAMP[crRNA]-based RNA manipulation tools. In addition, we present the structural basis of the functional relationship between the type III-E CRISPR-gRAMP effector complex and the caspase-like TPR-CHAT protease. Moreover, we demonstrated that the TPR-CHAT protease of gRAMP[crRNA]-TPR-CHAT Craspase complex cleaved Csx30 upon non-self-target RNA binding, which might be involved in the following antiphage defense. Furthermore, we provide a mechanism for self versus non-self RNA target discrimination governed by conformational

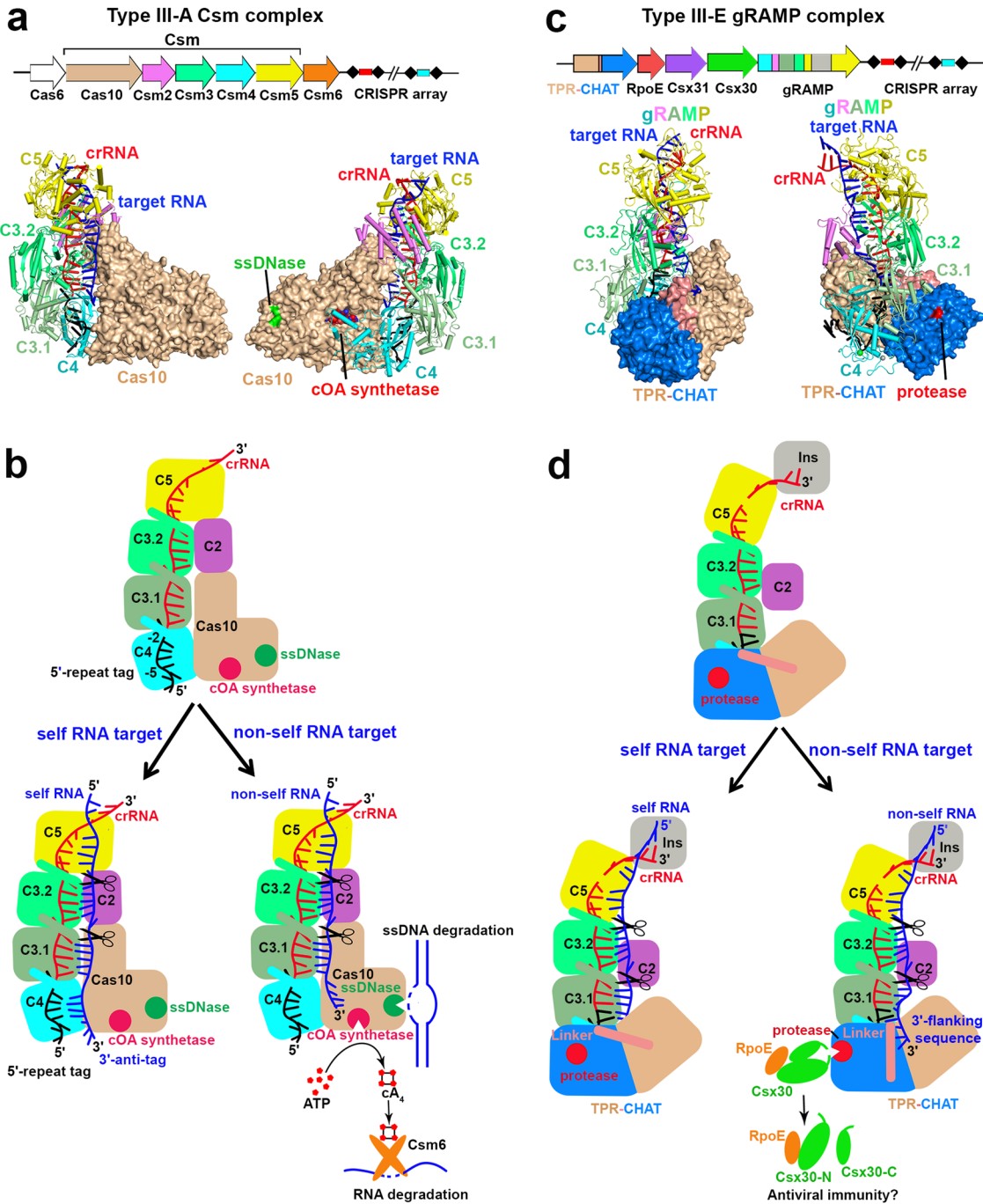

**Fig. 6 | Structural and mechanistic comparisons between type III-A and type III-E effector complexes.** Overall architecture of the type III-A (PDB 6MUR) (**a**) and III-E (**c**) effector complex and CRISPR-loci. **b** Model of type III-A interference during the immune defense. The type III-A muti-subunit effector complex recognizes the invasive non-self RNA targets with 3′-flanking sequence noncomplementary to the 5′-repeat tag, triggering the non-specific ssDNA cleavage and cOA synthesis by Cas10 protein. The produced cOA activates the Csm6 RNase activity, which is crucial for type III-A immunity. The Cas10 enzymic activities are switched off upon the subsequent target RNA cleavage by Csm3 subunits. Binding of the self RNA containing an 3′-anti-tag sequence complementary to the 5′-repeat tag of crRNA could not activate Cas10 activities, but still could be cleaved by Csm3 subunits.

**d** Model of type III-E interference during anti-phages or -plasmids infection. The type III-E single-protein effector directly interacts with a caspase-like protease TPR-CHAT, which adopts an autoinhibitory state. Binding of non-self RNA target activates the protease activity of the CHAT domain, leading to cleavage of target Csx30 protein, which forms a stable complex with RpoE. The resulting two products Csx30-C and RpoE-Csx30-N might be involved in the following antiphage defense. Cleavage of target RNA by Csm3 domains would shut down the protease activity. The 3′-anti-tag sequence takes a different binding route, leaving the protease to remain in an autoinhibitory inactive state. The bound self RNA targets could still be cleaved by Csm3 domains.

change-induced activation of the TPR-CHAT protease in type III-E CRISPR-Cas antiviral immunity. While our manuscript was under review, several studies[42–47] reported the structural and functional insights of type III-E Craspase complexes; these independent studies are overall in agreement with our results.

## Methods

### Protein expression and purification

The full-length *Candidatus* "Scalindua brodae" genes encoding gRAMP, TPR-CHAT, RpoE, Csx30, and Csx31 were synthesized by Sangon Biotech (Shanghai). The *csx30* gene was cloned into the

pCDFDuet-1 (Novagen, streptomycin resistance) vector with an N-terminal hexahistidine tag and a C-terminal Strep II tag. The *csx31* gene was cloned into a modified pET28a vector with a SUMO tag following a ubiquitin-like protease (ULP1) cleavable site and an MBP tag to increase the soluble protein yield, fused with an N-terminal hexahistidine tag. For the expression of Csx30 -RpoE complex, the *csx30* gene with an N-terminal hexahistidine tag and the *rpoE* gene with a C-terminal Strep II tag were subcloned into the two multiple cloning sites of pCDFDuet-1 vector, respectively. The gRAMP gene and the synthetic CRISPR gene were cloned into the pRSFDuet-1 vector (Novagen, kanamycin resistance), in which an N- terminal hexahistidine tag was fused with gRAMP. The TPR-CHAT gene was subcloned into the pETDuet-1 vector (Novagen, ampicillin resistance) with a C-terminal Strep II tag. Individual vector or two vectors were transformed into *Escherichia coli* BL21-CodonPlus (DE3). Streptomycin-resistant colonies for expression of Csx30 and Csx30-RpoE complex, kanamycin-resistant colonies for expression of Csx31, double-resistant colonies for expression of gRAMP$^{crRNA}$-TPR-CHAT complex or kanamycin-resistant colonies for expression of gRAMP$^{crRNA}$ complex were picked and grown to $OD_{600}$ of -0.8 in LB medium at 37 °C. The expression was induced by adding isopropyl-β-D-1-thiogalactopyranoside to 0.5 mM and shifted to at 16 °C for 20 h. Cells were harvested by centrifugation and resuspended in lysis buffer (20 mM Tris-HCl, 300 mM NaCl, 20 mM imidazole, 1 mM DTT, pH 8.0). The harvested cells were then lysed by the EmulsiFlex-C3 homogenizer (Avestin) and centrifuged at 20,000 rpm for 30 min.

For the purification of gRAMP$^{crRNA}$ complex or Csx31, the supernatant was applied to 5 mL HisTrap Fast flow column (Cytiva). The protein was eluted with lysis buffer supplemented with 500 mM imidazole after washing the column with 10 column volumes of lysis buffer and 2 column volumes of lysis buffer supplemented with 40 mM imidazole.

For the purification of the gRAMP$^{crRNA}$-TPR-CHAT complex, Csx30 or Csx30-RpoE complex, the elution from HisTrap Fast flow column was subjected to further purification by loading into 5 mL StrepTrap HP column (Cytiva). Proteins were eluted in buffer A (20 mM Tris-HCl, 100 mM NaCl, 1 mM DTT, pH 8.0) supplemented with 5 mM desthiobiotin and then loaded into 5 mL HiTrap Q Fast Flow column (Cytiva). Proteins were eluted in a linear gradient from 100 mM to 1 M NaCl in 20 column volumes, and then concentrated using 100 kDa molecular mass cut-off concentrators (Amicon) before final purification on a Superdex 200 increase 10/300 GL column (Cytiva) pre-equilibrated in buffer B (20 mM Tris-HCl, 150 mM NaCl, 1 mM DTT, pH 8.0). Pooled fractions were concentrated, flash-frozen in liquid nitrogen, and stored at −80 °C for further use.

All mutants were generated by site-directed mutagenesis and purified by the same method as above.

## In vitro assembly of gRAMP$^{crRNA}$ or gRAMP$^{crRNA}$-TPR-CHAT in complex with self or non-self RNA targets

To assemble gRAMP$^{crRNA}$-target RNA or gRAMP$^{crRNA}$-TPR-CHAT-target RNA ternary complex, the purified gRAMP$^{crRNA}$ binary or gRAMP$^{crRNA}$-TPR-CHAT ternary complex were mixed with either self RNA (5′ CUCUAGUAACAGCCGUGGAGUCCGGGGCAGAAAAUUGGGUACCGU GACAUUAAGUC 3′) or non-self RNA (5′ CUCUAGUAACAGCCGUGGA GUCCGGGGCAGAAAAUUGGCAUGGCACUGUAAUUCAG 3′) at a molar ration of 1:1.2 and then incubated on ice for 30 min. We have added 5 mM EDTA to chelate divalent cations and thus prevent target cleavage. The mixture was loaded on a Superdex 200 Increase 10/300 GL(Cytiva) column equilibrated with buffer B. Pooled fractions were concentrated, flash-frozen in liquid nitrogen, and stored at −80 °C.

## Cryo-EM sample preparation and data acquisition

3.5 μL of 2 mg/mL purified gRAMP$^{crRNA}$, gRAMP$^{crRNA}$-target RNA$^{self}$, gRAMP$^{crRNA}$-target RNA$^{non-self}$, gRAMP$^{crRNA}$-TPR-CHAT, gRAMP$^{crRNA}$-TPR-

CHAT-target RNA$^{self}$, and gRAMP$^{crRNA}$-TPR-CHAT-target RNA$^{non-self}$ complexes were applied onto glow-discharged UltrAuFoil 300 mesh R1.2/1.3 grids (Quantifoil), respectively. Grids were blotted for 2 s at 100% humidity, 4 °C, and flash frozen into liquid ethane using Vitrobot Mark IV (FEI). Images were collected with SerialEM v3.8.2 on the FEI Titan Krios electron microscope at acceleration voltage of 300 kV with a Gatan K3 Summit detector with a physical pixel size of 1.1 Å at a defocus range from −1.5 μm to −2.5 μm. Each movie comprises 32 subframes with a total dose of 50 e⁻/Å².

## Cryo-EM data processing

Image processing was performed by RELION 3.1[48] and cryoSPARC v3.1[49]. The motion correction was performed with MotionCor2[50]. Contrast transfer function (CTF) parameters were estimated by Ctffind4[51]. Auto-picked particles using Laplacian-of-Gaussian were extracted and subjected to two rounds of 2D classification and two rounds of 3D classification in cryoSPARC v3.1, using the initial model generated in cryoSPARC v3.1 as a reference. Particles corresponding to the best class with the highest-resolution features were selected and subjected to non-uniform refinement in cryoSPARC v3.1. The final particles were subjected to the CTF refinement and then subjected to non-uniform refinement, generating the final reconstitutions. To further improve the density of the Csm5 insertion domain in the gRAMP$^{crRNA}$-target RNA$^{self}$ complex, a soft mask for the insertion domain was generated and applied for the subsequent local refinement in cryoSPARC v3.1. The final map of gRAMP$^{crRNA}$-target RNA$^{self}$ complex was obtained by combining the maps from non-uniform refinement and local refinement in Chimera v2.4.2[52]. All resolutions were estimated by applying a soft mask around the protein density and the Fourier shell correlation (FSC) = 0.143 criterion in cryoSPARC v3.1. Local resolution estimates were calculated from two-half data maps in cryoSPARC v3.1. Further details related to data processing and refinement are summarized in Supplementary Table 1.

## Atomic model building and refinement

Atomic models were built de novo, then interactively refined in COOT v0.9.5[53]. All models were refined against the cryo-EM maps using phenix.real_space_refine v1.19.2[54] by applying geometric and secondary structure restraints. All figures were prepared in PyMol v2.4.2 (http://www.pymol.org) and Chimera v1.15[52]. The statistics for data collection and model refinement are shown in Supplementary Table 1.

## RNA cleavage assay

In vitro RNA cleavage assays were performed by incubating gRAMP$^{crRNA}$ complex or its mutants in different concentrations as indicated with 50 nM 5′ 6-FAM labeled RNA in 10 μL reaction buffer composed of 20 mM HEPES, 75 mM NaCl, 5 mM MgCl₂, 100 μM CoCl₂, pH 7.5 at 37 °C for 60 min. The reactions were quenched by the addition of protease K and 5× stop buffer (0.5 mg/mL bovine serum albumin, 5% SDS, 50 mM EDTA) followed by incubation for 10 mins at 95 °C. The samples were separated on 15% 8 M urea-PAGE and the FAM fluorescence signal was quantified with Canon 6100 Multi imaging system (Tanon Science & Technology Co., Ltd., Shanghai, China). The cleavage assays were repeated at least three times, and representative results are shown. For the presentation of full scan blots, please see the Source Data files and Supplementary Information files.

## Protease cleavage assay

The cleavage reactions were assembled in 10 μL reaction volume with buffer containing 20 mM Tris-HCl, 100 mM NaCl, pH 8.0. 1 μM gRAMP$^{crRNA}$-TPR-CHAT complex or its mutants (gRAMP_D547A _D698A$^{crRNA}$-TPR-CHAT, gRAMP$^{crRNA}$-TPR-CHAT_C627A, or gRAMP$^{crRNA}$-TPR-CHAT_C627S) were added to reactions with 5 μM proposed substrates (Csx30-RpoE, Csx30 or SUMO-MBP-Csx31), followed by the addition of self RNA or non-self RNA target. Reactions were incubated

at 37 °C for 60 min, then separated on 4–20% FuturePAGE™ protein gel (ACE). Gels were stained with Coomassie G-250, and imaged using ChampGel7000 (Sage).

The cleavage reaction of Csx30-RpoE complex was performed by gRAMP_D547A _D698A$^{crRNA}$-TPR-CHAT in presence of non-self RNA target in buffer containing 20 mM Tris-HCl, 100 mM NaCl, pH 8.0. Reaction was incubated at room temperature overnight. Then enzyme-digested products were purified by 1 mL HisTrap HP column (Cytiva), and the proteins were eluted in the lysis buffer supplemented with 300 mM imidazole, and the elutions were further purified by applying on Superdex 200 increase 10/300 GL column (Cytiva) equilibrated with buffer B. For the presentation of full scan blots, please see the Source Data files and Supplementary Information files.

### Reporting summary

Further information on research design is available in the Nature Portfolio Reporting Summary linked to this article.

## Data availability

The data that support this study are available from the corresponding authors upon request. The cryo-EM density maps and atomic coordinates generated in this study have been deposited in the Electron Microscopy Data Bank (EMDB) and the Protein Data Bank (PDB). The cryo-EM density maps have been deposited in EMDB under accession number EMD-33676 (gRAMP$^{crRNA}$); EMD-33678 (gRAMP$^{crRNA}$-target RNA$^{self}$); EMD-33677 (gRAMP$^{crRNA}$-target RNA$^{non-self}$); EMD-33680 (gRAMP$^{crRNA}$-TPR-CHAT); EMD-33681 (gRAMP$^{crRNA}$-TPR-CHAT-target RNA$^{self}$) and EMD-33679 (gRAMP$^{crRNA}$-TPR-CHAT-target RNA$^{non-self}$). The atomic coordinates have been deposited in the PDB with accession numbers 7Y80 (gRAMP$^{crRNA}$); 7Y82 (gRAMP$^{crRNA}$-target RNA$^{self}$); 7Y81 (gRAMP$^{crRNA}$-target RNA$^{non-self}$); 7Y84 (gRAMP$^{crRNA}$-TPR-CHAT); 7Y85 (gRAMP$^{crRNA}$-TPR-CHAT-target RNA$^{self}$) and 7Y83 (gRAMP$^{crRNA}$-TPR-CHAT-target RNA$^{non-self}$). Source data are provided with this paper.

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

## Acknowledgements

We thank Dinshaw J. Patel for critical reading and editing, and Jie Wang for helpful discussion on target RNA cleavage mechanisms. We thank the staff at Southern University of Science and Technology (SUSTech) Cryo-EM Center for assistance in data collection on the SUSTech Titan Krios cryo-electron microscope. This work was supported by the National Natural Science Foundation of China (Grant No. 32270050 to N.J.), Shenzhen and Guangdong Natural Science Foundation (Grant No. 202205303000517 and 2314050005743 to N.J.), the Shenzhen Government "Peacock Plan" (Y01416126 to N.J.), the Guangdong Provincial Science and Technology Innovation Council Grant (2017B030301018), the Shenzhen Science and Technology Program (KQTD20190929173906742), Key Laboratory of Molecular Design for Plant Cell Factory of Guangdong Higher Education Institutes (2019KSYS006), and the Shenzhen Government "Peacock Plan" (Y01226136 to H.H.).

## Author contributions

N.C. undertook biochemical and structural studies, from sample preparation and purification, biochemical assays, and cryo-EM data collection. J.T.Z. performed data processing, and structure refinement and analysis. In addition, C.W. shared his cryo-EM expertise with J.T.Z. and N.J. to facilitate the research. Z.L. and X.Y.L. contributed to recombinant plasmid construction and protein purification. N.J. and H.H. directed the research. N.J. wrote the manuscript with input from other authors.

## Competing interests

The authors declare no competing interests.
