## [Peer Review File · Nature Communications]

Structural basis for the non-self RNA-activated protease activity of the type III-E CRISPR nuclease-protease CraspaseEditorial Note: This manuscript has been previously reviewed at another journal that is not operating a transparent peer review scheme. This document only contains reviewer comments and rebuttal letters for versions considered at *Nature Communications*.

REVIEWERS' COMMENTS

Reviewer #1 (Remarks to the Author):

The revised manuscript from the Jia group is substantially improved. The new data pertaining to Csx30, and the increased transparency regarding map quality greatly strengthens the already strong manuscript. The mechanistic insights presented here will undoubtedly be of great interest to the field, and I recommend the publication of this manuscript without delay.

Jack Bravo

Reviewer #2 (Remarks to the Author):

The authors have addressed all my major concerns and most of my minor concerns. I just had one minor point. EDTA can chelate divalent cations including both Mg²⁺ and Zn²⁺. If the authors introduced EDTA in their cryo-EM samples, they need to justify the existence of Zn²⁺ in their models.

Reviewer #1 (Remarks to the Author):

The revised manuscript from the Jia group is substantially improved. The new data pertaining to Csx30, and the increased transparency regarding map quality greatly strengthens the already strong manuscript. The mechanistic insights presented here will undoubtedly be of great interest to the field, and I recommend the publication of this manuscript without delay.

Jack Bravo

We appreciate the reviewer's high evaluation and support of our work.

Reviewer #2 (Remarks to the Author):

The authors have addressed all my major concerns and most of my minor concerns. I just had one minor point. EDTA can chelate divalent cations including both Mg^{2+} and Zn^{2+} . If the authors introduced EDTA in their cryo-EM samples, they need to justify the existence of Zn^{2+} in their models.

We thank the reviewer for the supportive comments.

We have further checked the density for Zn^{2+} , which is in our map with traceable density (see the figure below). We speculated that the affinity of Zn^{2+} is so high that EDTA could not chelate it from its binding residues.

The cryoEM density of Zn^{2+} and its interacting residues in C3.2 domain of $gRAMP^{crRNA}$ -target RNA complex.